# A tRNA modification in *Mycobacterium tuberculosis* facilitates optimal intracellular growth

Francesca G Tomasi[1†], Satoshi Kimura[2,3,4*†‡], Eric J Rubin[1], Matthew K Waldor[1,2,3,4]

[1]Department of Immunology and Infectious Diseases Harvard T. H. Chan School of Public Health, Boston, United States; [2]Division of Infectious Diseases, Brigham and Women's Hospital, Boston, United States; [3]Department of Microbiology, Harvard Medical School, Boston, United States; [4]Howard Hughes Medical Institute, Boston, United States

*For correspondence: sk2956@cornell.edu

†These authors contributed equally to this work

Present address: ‡Department of Microbiology and Immunology, Cornell University, Ithaca, United States

Competing interest: The authors declare that no competing interests exist.

**Abstract** Diverse chemical modifications fine-tune the function and metabolism of tRNA. Although tRNA modification is universal in all kingdoms of life, profiles of modifications, their functions, and physiological roles have not been elucidated in most organisms including the human pathogen, *Mycobacterium tuberculosis* (*Mtb*), the causative agent of tuberculosis. To identify physiologically important modifications, we surveyed the tRNA of *Mtb*, using tRNA sequencing (tRNA-seq) and genome-mining. Homology searches identified 23 candidate tRNA modifying enzymes that are predicted to create 16 tRNA modifications across all tRNA species. Reverse transcription-derived error signatures in tRNA-seq predicted the sites and presence of nine modifications. Several chemical treatments prior to tRNA-seq expanded the number of predictable modifications. Deletion of *Mtb* genes encoding two modifying enzymes, TruB and MnmA, eliminated their respective tRNA modifications, validating the presence of modified sites in tRNA species. Furthermore, the absence of *mnmA* attenuated *Mtb* growth in macrophages, suggesting that MnmA-dependent tRNA uridine sulfation contributes to *Mtb* intracellular growth. Our results lay the foundation for unveiling the roles of tRNA modifications in *Mtb* pathogenesis and developing new therapeutics against tuberculosis.

## eLife assessment

This is a **valuable** addition to the literature as it helps us understand the role of tRNA modifying enzymes in Mycobacterium tuberculosis. By knocking out one of the enzymes, the authors **convincingly** demonstrate the importance of tRNA-modifying enzymes for intra-host growth of tubercle bacteria. Some of the claims regarding modification as well as the role in virulence could be strengthened through further bioinformatics and phylogenetic analyses as well as experimental approaches. The work will be of interest to microbiologists.

## Introduction

tRNA is an adaptor molecule that enables protein synthesis by converting the triplet genetic code in mRNA into amino acids. The fidelity of base pairing between mRNA codons and tRNA anticodons is monitored within ribosomes and is critical for properly incorporating the amino acids bound to the 3′ ends of tRNAs into growing polypeptides. For optimal translation, the abundances, and properties of tRNA isoacceptors are fine tuned by diverse mechanisms, including chemical modifications (*Huang and Hopper, 2016*; *Björk and Hagervall, 2014*; *Shepherd and Ibba, 2015*; *Torrent et al.,*

*2018*). Dysregulation of tRNA abundance and/or structure leads to defective decoding and results in ribosome pausing and collisions, protein misfolding, stress responses and can have detrimental or lethal effects on the cell (*Orellana et al., 2022*; *Suzuki, 2021*; *Liu et al., 2022*; *Goodarzi et al., 2016*; *Delaunay et al., 2022*; *Nedialkova and Leidel, 2015*).

Chemical modifications of tRNA (tRNA modifications) are found in all kingdoms of life and fine-tune tRNA properties including mRNA decoding efficiency, recognition by aminoacyl-tRNA synthetases, half-life, and structural stability (*Björk and Hagervall, 2014*; *Nedialkova and Leidel, 2015*; *Kimura and Waldor, 2019*; *Giegé and Eriani, 2023*). Modifications are prevalent in the anticodon loop, particularly at the first letter of the anticodon. Modifications of the anticodon loop directly modulate codon recognition, whereas modifications in the tRNA body region primarily stabilize tRNA tertiary structure, protecting them from degradation in the cell. tRNA modifications are generated by dedicated site-specific enzymes referred to as tRNA modifying enzymes. tRNA modifications have been extensively characterized in a few model organisms (*Björk and Hagervall, 2014*; *de Crécy-Lagard and Jaroch, 2021*), but their profiles, regulation, and functions in non-model organisms, including bacterial pathogens, are understudied (*de Crécy-Lagard and Jaroch, 2021*).

tRNA sequencing (tRNA-seq) allows for the rapid and systematic prediction of many tRNA modification sites (*Zhang et al., 2022*; *Zheng et al., 2015*). We recently developed a comparative tRNA-seq protocol to profile tRNA modifications in organisms with uncharted tRNA modification profiles; in *Vibrio cholerae*, this approach led to the discovery of a new RNA modification and RNA editing process (*Kimura et al., 2020*). tRNA-seq enables rapid prediction of modified sites through detection of reverse transcription-derived signatures, such as nucleotide misincorporation and early termination, both of which occur more frequently at modified sites. Furthermore, several chemical treatments of tRNA can convert modifications that are not recognizable as reverse transcription-derived signatures into detectable signals, expanding the repertoire of modifications that can be distinguished by tRNA-seq (*Motorin and Helm, 2019*; *Finet et al., 2022*; *Draycott et al., 2022*; *Dai et al., 2023*).

*Mycobacterium tuberculosis* (*Mtb*), the agent of tuberculosis (TB), is a global pathogen that caused >10.5 million cases and over 1.5 million deaths worldwide in 2020 (*W.H.O, 2021*). Several studies have uncovered roles for non-*Mtb* mycobacterial tRNA modifications in stress responses, adaptation to environmental changes, and persister cell formation (*Chionh et al., 2016*). *Mycobacterium bovis* BCG, an organism closely related to *Mtb*, responds to hypoxia by reprogramming 40 ribonucleoside modifications in tRNA to facilitate translation of a subset of proteins that promote survival in hypoxic conditions (*Chionh et al., 2016*). To date, studies of the profiles and functions of *Mtb* tRNA modifications have been limited.

Here, we conducted tRNA-seq in *Mtb*. We assigned modifications to many of the reverse transcription-derived signatures identified, using information on the presence of homologs of known modifying enzymes in *Mtb*. Chemical treatments of tRNAs carried out prior to tRNA-seq increased the detectability of certain modifications. We constructed two *Mtb* deletion mutant strains, with deletions of *mnmA* and *truB*, and confirmed that the absence of these modification enzymes eliminated the predicted signals in tRNA-seq data. Furthermore, while deletion of *mnmA* in *Mtb* did not affect the pathogen's growth in in vitro laboratory growth conditions, the *mnmA* knockout strain's growth was attenuated in a macrophage infection model. Our findings suggest that tRNA modifications warrant further study as we unravel the complexity of *Mtb* infections, as they may serve as targets for new therapeutics.

## Results

### In silico prediction of *Mtb* tRNA modifying enzymes

To predict tRNA modifications in *Mtb*, we used Basic Local Alignment Search Tool (BLAST) (*Altschul et al., 1990*) to identify homologs of all tRNA modification enzymes registered in Modomics in the *Mtb* genome (*Boccaletto et al., 2022*; *Supplementary file 1*). With a stringent threshold (*E*-value <1 × 10$^{-10}$), 31 *Mtb* genes homologous to genes encoding known RNA modification enzymes were identified (*Supplementary file 2*). Twenty-three of these genes are predicted to synthesize 16 tRNA modifications in *Mtb* (*Supplementary files 2 and 3*), including *miaA* and *miaB* for 2-methylthio-6-isopentenyl-adenosine (ms$^2$i$^6$A), *tsaD*, *tsaB*, *tsaE*, and *sua5* for $N^6$-threonylcarbamoyladenosine (t$^6$A), *mnmA* for 2-thiouridine (s$^2$U), *truB*, *truA*, *rluA*, and *pus9* for pseudouridine (Ψ), *trm2* for 5-methyluridine (m$^5$U),

*trmD* for 1-methylguanosine (m$^1$G), *trmI* for 1-methyladenosine (m$^1$A), *trmL* for 2′-*O*-methylcytidine (Cm) or 2′-*O*-methyluridine (Um), *trmB* for 7-methylguanosine (m$^7$G), two *trmH* for 2′-*O*-methylguanosine (Gm), *trmR* for 5-methoxyuridine (mo$^5$U), *trcM* for 5-methylcytidine (m$^5$C), *dusB* for dihydrouridine (D), *tadA* for inosine (I), and *tilS* for lysidine (k$^2$C). While eight additional genes met the threshold for homology to modification enzymes, they exhibited greater similarity to a Glutamyl-tRNA synthase; Rv2992c, ribosome associated GTPases (Der; Rv1713 and Era; Rv2364c), molybdopterin biosynthesis proteins (MoeW; Rv2338c and MoeB2; Rv3116), thiosulfate sulfur transferases (CysA; Rv3117 and SseA; Rv3283) and a riboflavin biosynthesis protein (RibG; Rv1409), and likely do not correspond to tRNA modification enzymes (*Supplementary file 2*).

We mined data from genome-wide Tn-seq and CRISPRi screens (*DeJesus et al., 2017*; *Bosch et al., 2021*) to assess the impacts of *Mtb* tRNA modifications on its growth. Five genes encoding *Mtb* tRNA modifying enzymes, *trmD, tilS, tadA, tsaC2,* and *miaA*, were reported to be essential for *Mtb* growth in both Tn-seq (*DeJesus et al., 2017*) and CRISPRi screens (*Bosch et al., 2021*; *Supplementary file 3*), suggesting that the modifications they produce are critical for tRNA functions. Indeed, *E. coli trmD, tilS, tadA, and tsaC2* are also essential for growth and critical for codon decoding, aminoacylation, and reading frame maintenance (*Masuda et al., 2022*; *Soma et al., 2003*; *Wolf et al., 2002*; *El Yacoubi et al., 2009*). The *Mtb* modifications synthesized by these enzymes likely have similar impacts on tRNA functions. Unexpectedly, one modifying enzyme, MiaA, is non-essential in *E. coli* grown in nutrient-rich medium, but apparently essential in *Mtb*, suggesting that i$^6$A, the modification introduced by MiaA, may have more profound roles in *Mtb* translation than in *E. coli*.

Several of the putative *Mtb* tRNA modifying enzymes are conserved across all three domains of life (e.g., TruB and TsaD) (*Figure 1*). By contrast, some enzymes were limited to bacterial species closely related to *Mtb*, possibly suggesting their species-specific physiological roles, including pathogenesis. For example, TrmI (Rv2118c) homologs are widely present in Archaea and Eukaryotes, but are sparsely distributed in bacteria, where they are primarily limited to Actinomycetia, including *Mtb*, and several thermophilic bacterial species (e.g., *Thermus thermophilus*). TrmI synthesizes m$^1$A at position 58 in eukaryotes (*Anderson et al., 1998*) and at position 57/58 in archaea (*Guelorget et al., 2010*); indeed, TrmI in *Mtb* has been proven to generate m$^1$A at position 58 (*Varshney et al., 2004*). Pus9 (Rv3300c) has many homologs exhibiting weak similarity across eukaryotes and bacteria. However, some species, including *Pseudomonas aeruginosa*, *Acinetobacter baumannii*, *Neisseria gonorrhoeae*, and several species of Actinomycetia, encode proteins showing strong homology to *Mtb* Pus9, suggesting that this set of pseudouridylases have a distinctive property such as substrate specificity.

tRNA modifying enzymes predicted in *Mtb* were observed in mycobacteria species, including *Mycolicibacterium smegmatis* and *Mycobacteroides abscessus* (*Figure 1*), suggesting that tRNA modification patterns are similar among mycobacterium species.

## Profiling *Mtb* tRNA modification sites by tRNA-seq

To begin profiling detectable *Mtb* tRNA modifications, we sequenced tRNAs isolated from wild-type *Mtb* strain H37Rv grown in 7H9 medium. In this protocol, tRNAs are first reversed transcribed to cDNA (*Kimura et al., 2020*). During cDNA synthesis, chemical modifications on tRNA nucleotides disrupt Watson–Crick base pairing and increase the frequency of reverse transcriptase errors, leading to incorporation of the incorrect nucleotide or early termination of cDNA synthesis (*Kellner et al., 2010*). These reverse transcription-derived 'signatures' typically correspond to modified sites (*Zhang et al., 2022*; *Zheng et al., 2015*) and are depicted in the heatmap in *Figure 2*.

Comparison of the reverse transcription-derived signatures observed in *Mtb* to *E. coli*, where tRNA modifications have been well characterized (*Kimura et al., 2020*) enables the prediction of the presence of common modifications, including ms$^2$i$^6$A, m$^1$G, I, and k$^2$C (*Figure 2* and *Figure 2—figure supplement 1*). These predictions are strongly supported by the set of tRNA modification enzymes identified in the *Mtb* genome (*Figure 1* and *Supplementary file 3*), including *miaA* and *miaB* (ms$^2$i$^6$A), *trmD* (m$^1$G), *tadA* (I), and *tilS* (k$^2$C). Some tRNA modifications were observed in *Mtb* but are not present in *E. coli*. In other actinobacteria, A58 and A59 are likely modified to m$^1$A (*Schwartz et al., 2018*), and since *trmI*, the methylase that generates this modification is present in *Mtb* (*Varshney et al., 2004*), most *Mtb* tRNAs likely contain this modification as well. Nucleoside variation in the sequence of tRNA genes can account for some of the variations in modified sites between *E. coli* and *Mtb*. For example, in *Mtb*, termination signatures derived from G at position 37 were detected

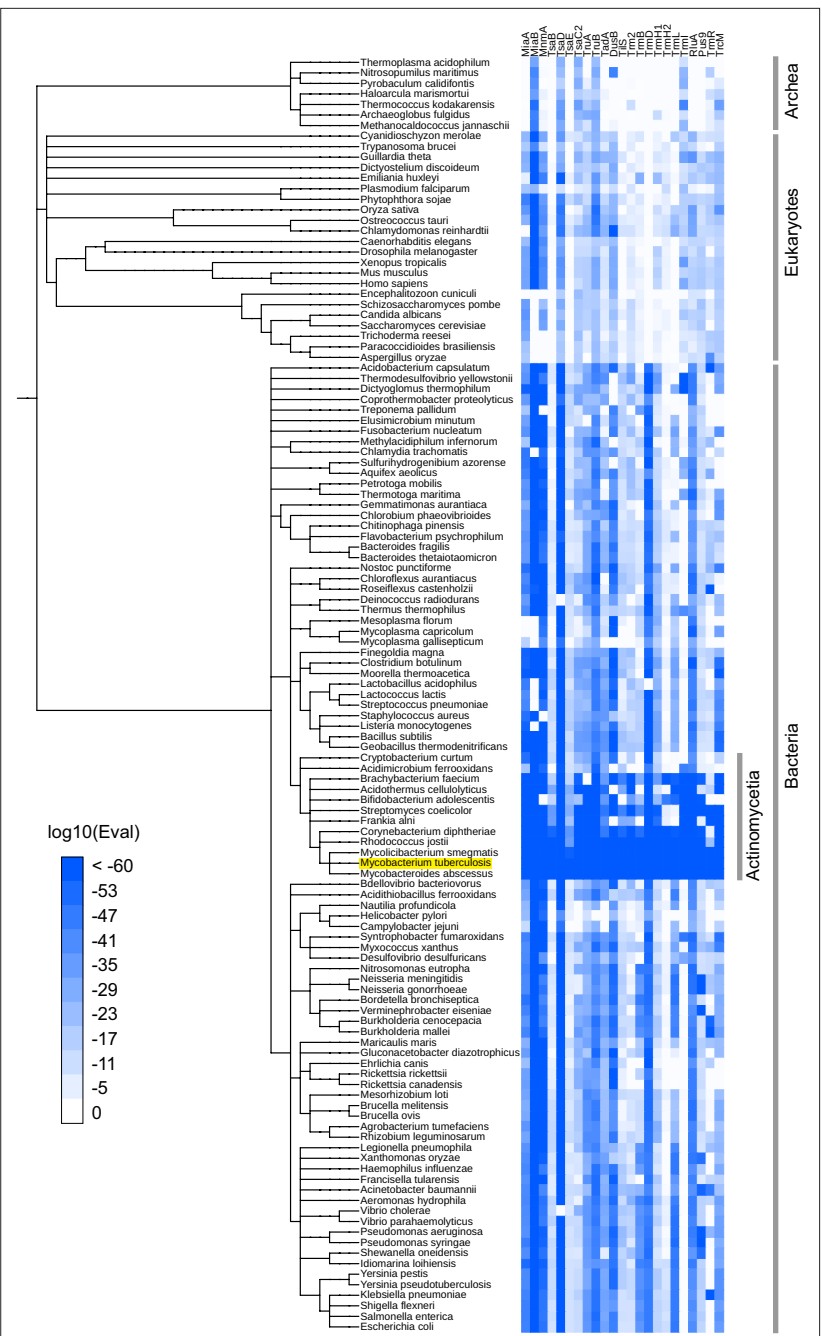

**Figure 1.** Phylogenetic distribution of *Mtb* tRNA modifying enzyme homologs. Heatmap of log10 *E*-values from BLAST search results. BLAST searches were conducted against 120 manually picked organisms using *Mtb* tRNA modifying enzymes as queries. When one organism has multiple hits, the lowest log10(Eval) values among hits are shown. iTol (**Letunic and Bork, 2021**) was used to depict the results.

The online version of this article includes the following source data for figure 1:

**Source data 1.** log10 *E*-values from BLAST searching for the homologs of *Mtb* tRNA modifying enzymes in 120 organisms.

in tRNA-Arg2, -Gln1, and -Gln2, whereas this position in these tRNAs in *E. coli* are modified A, such as $m^2A$ and $m^6A$, which are silent in tRNA-seq. Since $m^1G$ induces strong termination during reverse transcription, these positions are likely modified to $m^1G$, as observed in the *Bacillus subtilis* tRNA-Arg2 gene position 37 G (**Jühling et al., 2009**).

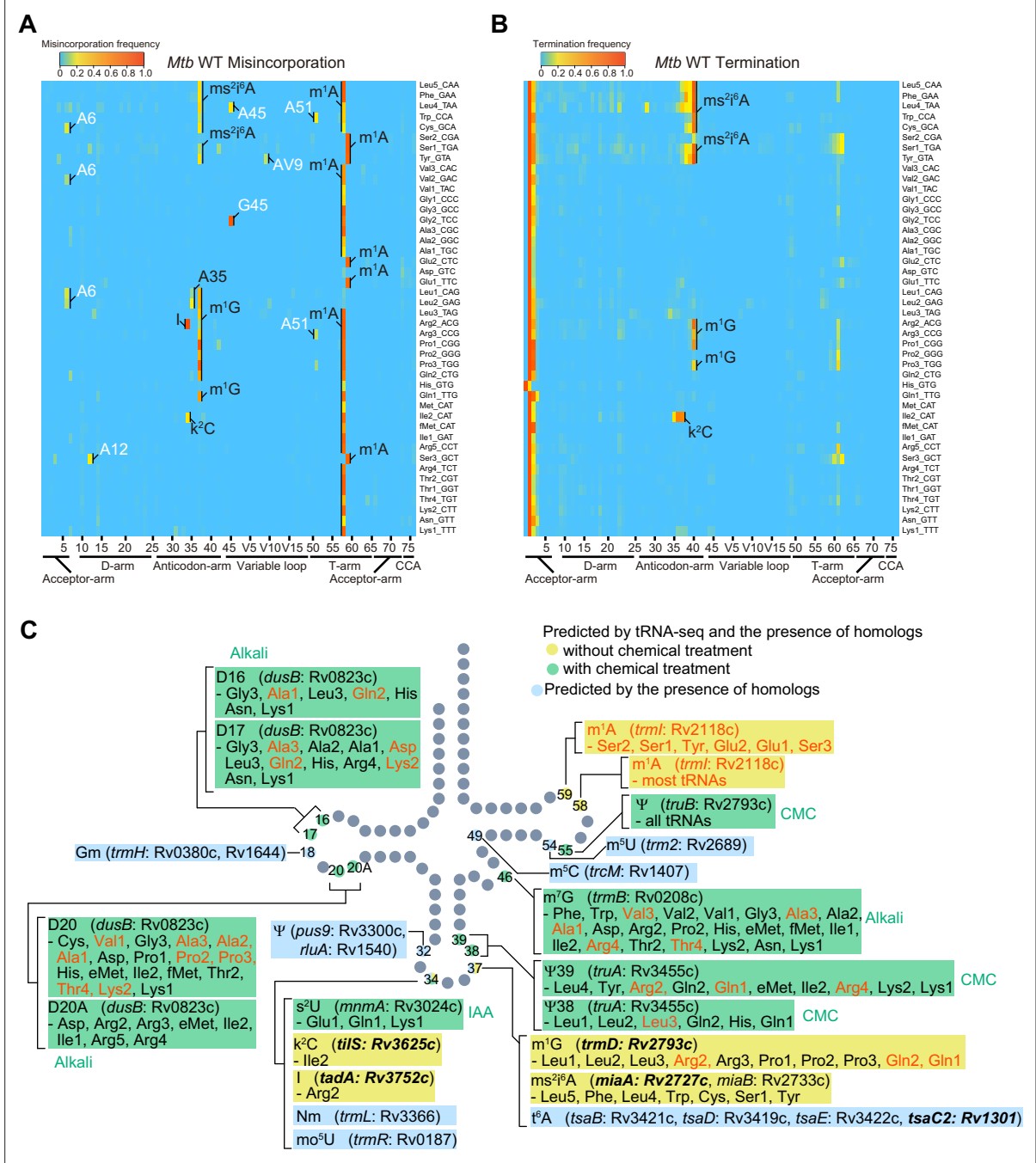

**Figure 2.** Heatmap of misincorporation and early termination frequency in sequencing of tRNAs from wild-ype *Mtb*. Heatmaps show misincorporation (**A**) and termination (**B**) frequencies at all positions across tRNAs (read 5' to 3'). Predicted modifications are labeled based on similarity to known modifications in other organisms and the presence of the tRNA modifying enzyme homologs (***Supplementary file 3***). The positions with more than 10% misincorporation in *Mtb* but not in *E. coli* are depicted in white in A. Representative data of two independent experiments with similar results are shown. (**C**) *M. tuberculosis* tRNA modifications predicted in this study. Schematic tRNA secondary structure with sites of modifications identified either by the presence of modifying enzymes and/or tRNA sequencing (tRNA-seq). Modifications and tRNA species that are not observed in *E. coli* are shown in red. Modifications and positions that are predicted by both RT-derived signature and the presence of the homologs of tRNA modifying enzymes are shown in yellow (without chemical treatment) and green (with chemical treatment), whereas modifications that are only predicted by the presence of the homologs are shown in light blue. Genes reported to be essential in *Mtb* are shown in bold.

The online version of this article includes the following source data and figure supplement(s) for figure 2:

**Source data 1.** Misincorporation and early termination frequencies in sequencing of tRNAs from wild-type *Mtb*.

**Figure supplement 1.** *E. coli* tRNA modifications detected by tRNA sequencing (tRNA-seq).

tRNA samples were also treated with several chemical treatments prior to sequencing, to expand the set of tRNA modifications detectable by tRNA-seq. These treatments included iodoacetamide (IAA), for detection of sulfur modifications, 1-cyclohexyl-(2-morpholinoethyl) carbodiimide (CMC) for detection of $\Psi$, and alkali for detection of D and $m^7G$. The chemical treatment protocols were first carried out with *E. coli*, to validate the methods.

IAA is a thiol-reactive compound that covalently attaches carboxyamidomethyl to thiolated uridines via nucleophilic substitution (*Herzog et al., 2017*) and modified $s^4U$ is detected as C instead of U. In IAA-treated samples, positions 8 and 9, corresponding to $s^4U$ in many tRNA species, had high misincorporation frequencies, confirming that IAA treatment modifies $s^4U$, leading to elevated misincorporation (*Figure 3—figure supplement 1*). Furthermore, we observed higher misincorporation and termination signals at the positions corresponding to other sulfur modification, $s^2C$, $s^2U$ and their derivatives, such as position 32 in tRNA-Arg3, -Arg5, -Ser3, and Arg4, and 34 in tRNA-Glu, -Gln1, and -Lys (*Figure 3* and *Figure 3—figure supplement 1*), revealing that IAA treatment facilitates the detection of not only $s^4U$ but also additional sulfur modifications, which are weakly detected without the IAA treatment.

Next, we applied IAA treatment to *Mtb* tRNA-seq. IAA treatment increased termination signals from position 34 in tRNA-Glu1, -Gln1, and -Lys, which contain $s^2U$ derivatives in *E. coli* (*Figure 4A*). The 2-thiouridine modification is carried out by MnmA in *E. coli* (*Kambampati and Lauhon, 2003*), and a homolog, *Rv3024c*, of this enzyme was identified in the *Mtb* genome (*Supplementary file 3*; *Kapopoulou et al., 2011*). We used double-stranded DNA-based recombineering to delete Rv3024c in *Mtb*, yielding strain *MtbΔmnmA* (*van Kessel and Hatfull, 2007*). Sequencing of tRNA isolated from *MtbΔmnmA* with prior IAA treatment showed reduced termination signals from position 34 in tRNA-Glu1, -Gln1, and -Lys in treated samples, indicating that Rv3024c plays a critical role in the modification responsible for increased termination frequency derived from position 34 in these tRNAs (*Figure 4B, C*). Together, these observations strongly suggest that Rv3024c encodes an MnmA-like enzyme that sulfurates position 34 uridines in three *Mtb* tRNA isoacceptors.

As shown previously (*Carlile et al., 2014*), CMC treatment increased both misincorporation and termination signatures at a subset of $\Psi$s in *E. coli* tRNAs (*Figure 5*, and *Figure 5—figure supplement 1* and *Figure 5—figure supplement 2*). In addition, CMC-treated samples showed increased frequencies of both misincorporation and termination at positions 16, 17, 20, and 20A corresponding to D, and $m^7G$ at position 46. Both modifications are known to undergo base elimination in mild alkali conditions (*Marchand, 2021*). Since these signals were also observed in the reaction condition in which CMC was not added, these signals are likely attributable to the alkali treatment that is common to both conditions.

Reverse transcription-derived signatures derived from alkali-treated D showed a distinctive pattern. With this treatment, when two Ds are at consecutive positions, for example, D16 and D17, termination signals were elevated at the following position, that is, position 18 (*Figure 5—figure supplement 1*, *Figure 5—figure supplement 3*, and *Figure 5—figure supplement 4*). Furthermore, alkali treatment also led to higher misincorporation frequencies at singlet Ds (*Figure 5*, *Figure 5—figure supplement 3*, and *Figure 5—figure supplement 4*). Thus, termination and misincorporation signatures enabled the prediction of known *E. coli* tRNA sites modified to D.

CMC/alkali treatment facilitated the identification of additional modifications in *Mtb* tRNAs. Regardless of CMC treatment, alkali-treated samples showed increased misincorporation and termination frequencies derived from U located at positions 16, 17, 20, and 20A (*Figure 6* and *Figure 6—figure supplement 1*). As observed in *E. coli*, termination signals at positions 18 and 21 likely correspond to consecutive Ds at positions 16 and 17, and 20 and 20A, respectively. *Mtb* Rv0823c is a homolog of dihydrouridylase DusB (*Supplementary file 3*), which likely accounts for the synthesis of D at these positions. Furthermore, alkali treatment increased the misincorporation frequencies at G at position 46 (*Figure 6—figure supplement 2*). Since Rv0208c is a homolog of TrmB, which synthesizes $m^7G$ at position 46 in *E. coli*, multiple *Mtb* tRNA species likely contain $m^7G$ at position 46 (*Figure 6* and *Figure 6—figure supplement 2*).

CMC treatment also increased the termination frequency at several sites. Termination signatures derived from position 55, which is exclusively uridine in all tRNA species, increased in most tRNA species, suggesting that *Mtb* tRNAs contain pseudouridines at this position (*Jühling et al., 2009*). Rv2793c is an *Mtb* homolog of *E. coli* TruB and deletion of Rv2793c reduced the termination

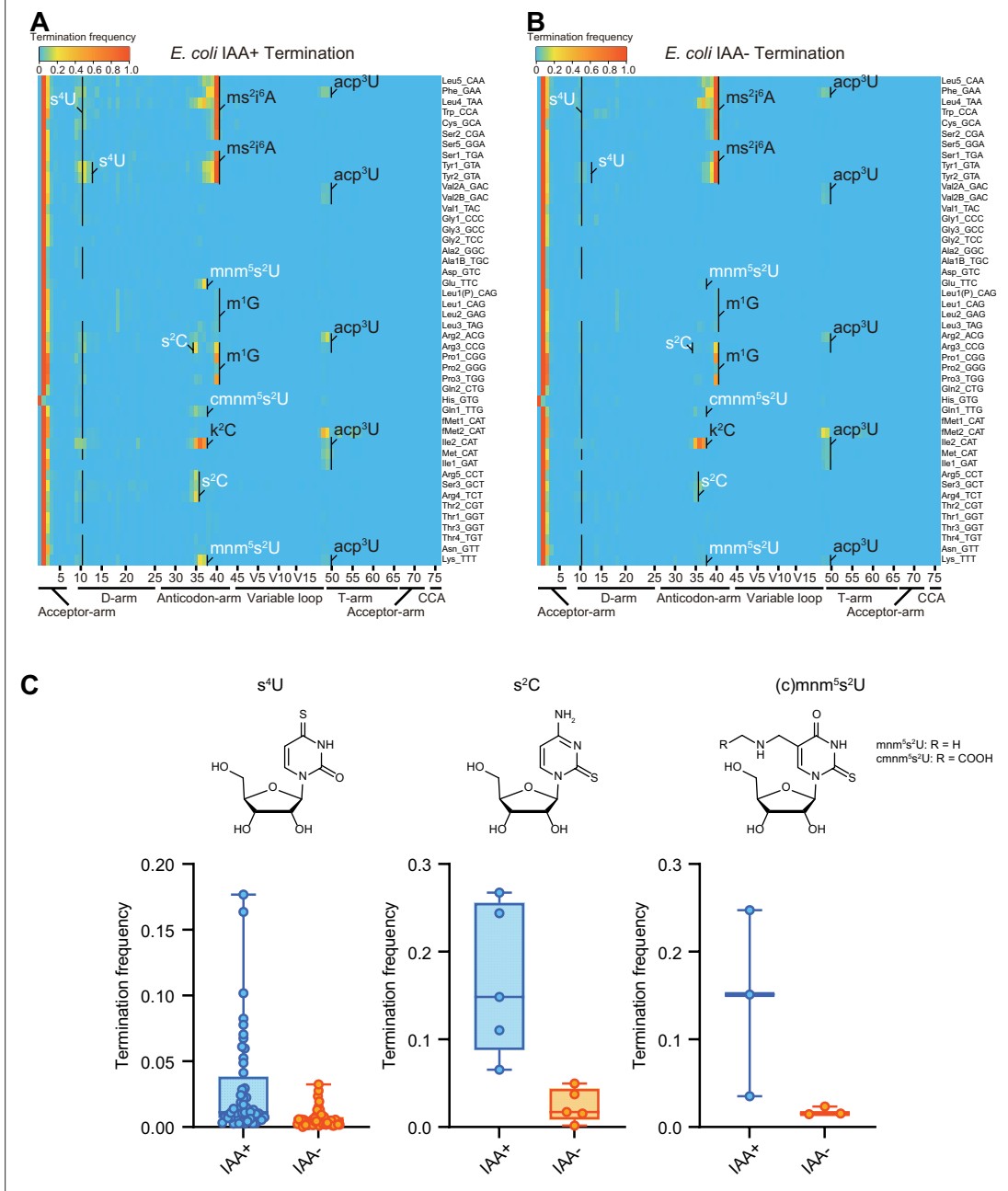

**Figure 3.** Iodoacetamide (IAA) treatment promotes detection of sulfur modifications on tRNAs by enhancing termination signals. Heatmaps of the termination signals of *E. coli* tRNAs treated with (**A**) or without (**B**) IAA. Known modification sites, including sulfur modifications (s⁴U, s²C, s²U in white) are shown. (**C**) Termination frequency at s⁴U (n = 48), s²C (n = 5), and s²U (n = 3) sites of tRNAs treated with or without IAA. The experiment was performed once.

The online version of this article includes the following source data and figure supplement(s) for figure 3:

**Source data 1.** Termination frequencies of *E. coli* tRNAs treated with or without iodoacetamide (IAA).

**Figure supplement 1.** Iodoacetamide (IAA) treatment promotes detecting sulfur modifications by enhancing misincorporation signals.

**Figure supplement 1—source data 1.** Misincorporation frequencies of *E. coli* tRNAs treated with or without iodoacetamide (IAA).

frequencies at this position in tRNAs isolated from *MtbΔtruB* (*Figure 6* and *Figure 6-figure supplement 3*; *van Kessel and Hatfull, 2007*). Together, these observations suggest that Rv2793c encodes a TruB-like enzyme that modifies position 55 uridines to Ψ across tRNA species. Furthermore, the presence of a TruA homolog in *Mtb* (Rv3455c) suggests that in multiple tRNA species U at positions

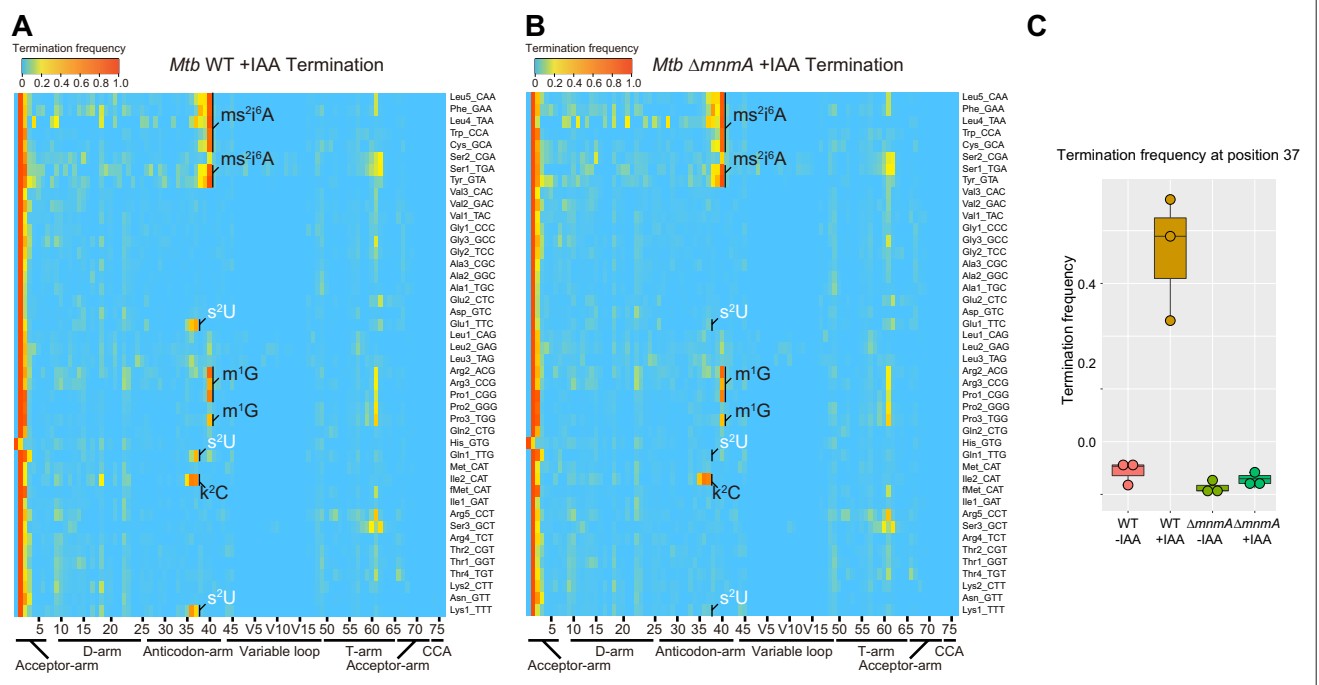

**Figure 4.** Heatmap and plot of early termination frequency from sequencing of tRNAs from wild-type and *MtbΔmnmA* with and without RNA alkylation. Heatmap of early termination frequencies across tRNA molecules and positions for wild-type (WT) (**A**) and *MtbΔmnmA* (**B**). Sulfur modification is shown in white. (**C**) Plot of termination frequencies at position 37 in WT *Mtb* and *MtbΔmnmA* for lysine_UUU, glutamate_UUG, and glutamine_UUC isoacceptors (n = 3). IAA: iodoacetamide. The experiment was performed once.

The online version of this article includes the following source data for figure 4:

**Source data 1.** Termination frequencies from sequencing of tRNAs from wild-type and *MtbΔmnmA* with and without RNA alkylation.

38–40 can be modified to Ψ. Indeed, the termination signatures derived from positions 38 and 39 increased depending on CMC treatment, strongly suggesting that these positions are modified to Ψ (*Figure 6* and *Figure 6—figure supplement 3*).

In total, among 16 tRNA *Mtb* tRNA modifications predicted by the presence of tRNA modifying enzymes (*Figure 1* and *Supplementary file 3*), 9 species of modifications were detected based on reverse-transcription-derived signatures (*Figure 2C*).

## Growth of *MtbΔmnmA* is attenuated in a macrophage infection model

To address whether *Mtb* tRNA modifications impact the pathogen's growth in the host environment, we used the MtbTnDB transposon insertion sequencing (Tn-seq) database (*Zhang et al., 2013*; *Jinich et al., 2021*) to determine if transposon insertions in genes encoding tRNA modifying enzymes have been associated with in vivo growth defects. Transposon insertions in *mnmA* were reported to attenuate *Mtb* growth in mice infected with a library of *Mtb* transposon mutants, suggesting that *mnmA* facilitates *Mtb* growth in vivo. We found that the growth of WT and *MtbΔmnmA* were similar in 7H9 medium (*Figure 7*), suggesting the absence of s²U modification at position 34 does not impair *Mtb* growth in culture. In contrast, the *MtbΔmnmA* mutant was significantly impaired for growth in a macrophage infection model (*Jinich et al., 2021*; *Figure 7*). Defective growth of the *MtbΔmnmA* mutant was also observed in macrophages treated with all-trans retinoic acid (ATRA), which promotes macrophage control of *Mtb* infection (*Babunovic et al., 2022*). These observations strongly suggest that modification of U to s²U by MnmA facilitates *Mtb* growth in macrophages.

## Discussion

Here, we profiled *Mtb* tRNA modifications with tRNA-seq to provide the first maps of the tRNA modification landscape in this global pathogen. In total, nine modifications, including six modifications

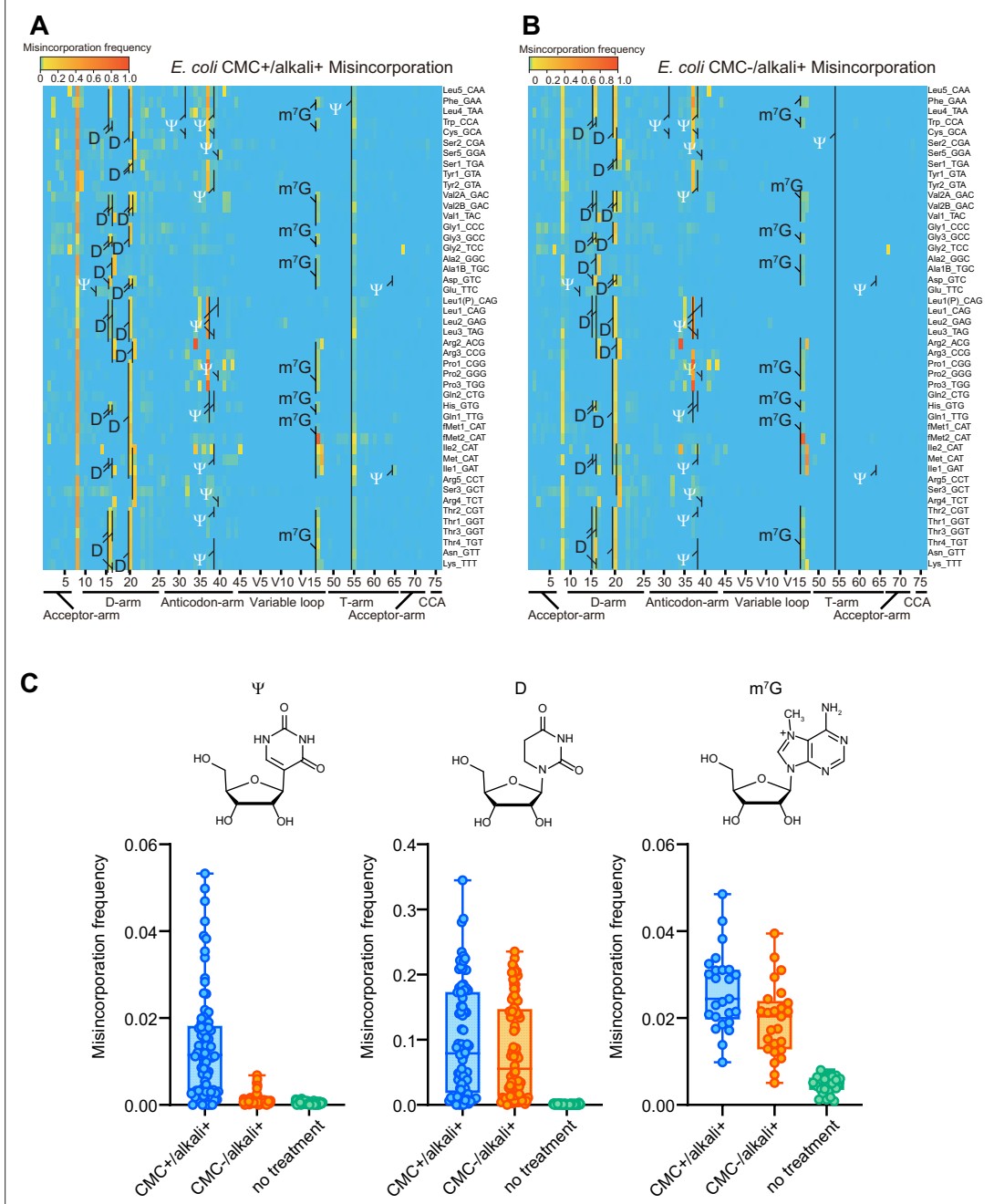

**Figure 5.** 1-Cyclohexyl-(2-morpholinoethyl) carbodiimide (CMC) and alkali treatment facilitate detection of $\phi$, D, and m$^7$G modifications in *E. coli*. Heatmaps of the misincorporation signals of *E. coli* tRNAs treated with (**A**) or without (**B**) CMC. In both conditions, tRNAs are incubated in alkali condition. Known $\Psi$, D, and m$^7$G sites are shown. $\Psi$ is shown in white. (**C**) Misincorporation frequency at known $\Psi$ (n = 80), D (n = 76), and m$^7$G (n = 25) sites of tRNAs treated with CMC+/alkali+ or CMC−/alkali+, and tRNAs without treatment. The experiment was performed once.

The online version of this article includes the following source data and figure supplement(s) for figure 5:

**Source data 1.** Misincorporation frequencies from sequencing of *E. coli* tRNAs treated with or without 1-cyclohexyl-(2-morpholinoethyl) carbodiimide (CMC).

**Figure supplement 1.** 1-Cyclohexyl-(2-morpholinoethyl) carbodiimide (CMC) and the following alkali treatment facilitate detecting additional modifications.

**Figure supplement 1—source data 1.** Termination frequencies from sequencing of *E. coli* tRNAs treated with or without 1-cyclohexyl-(2-morpholinoethyl) carbodiimide (CMC).

**Figure supplement 2.** 1-Cyclohexyl-(2-morpholinoethyl) carbodiimide (CMC) treatment facilitate detecting $\phi$.

*Figure 5 continued on next page*

*Figure 5 continued*

**Figure supplement 2—source data 1.** Termination frequencies at the uridine in *E. coli* tRNAs at positions 40, 41, and 57.

**Figure supplement 3.** Misincorporation and termination signals derived from D in *E. coli* tRNAs.

**Figure supplement 3—source data 1.** Termination frequencies at positions 18, 21, and 22, and misincorporation frequencies at positions 16, 17, 20, 20A, and 21 to predict the presence of D in *E. coli* tRNAs.

**Figure supplement 4.** RT-derived signatures originated from tandem Ds or singlet Ds.

without chemical treatment, were identified based on reverse transcription-derived signatures. CMC/alkali treatment and IAA treatment further identified $\Psi$ and m[7]G and sulfur modifications, respectively. Although we did not chemically validate the modifications predicted by tRNA-seq with mass spectrometry, the identification of *Mtb* homologs of tRNA modifying enzymes strongly bolsters the RT-signature-based predictions. Furthermore, the deletion of *truB* and *mnmA* genes in *Mtb* eliminated the respective modification signatures of pseudouridine and s[2]U, validating that the enzymes encoded by these genes synthesize these modifications. Finally, the growth defect of the *ΔmnmA* strain within macrophages but not in a nutrient-rich medium suggests that s[2]U tRNA modification facilitates *Mtb* adaptation to the host intracellular environment.

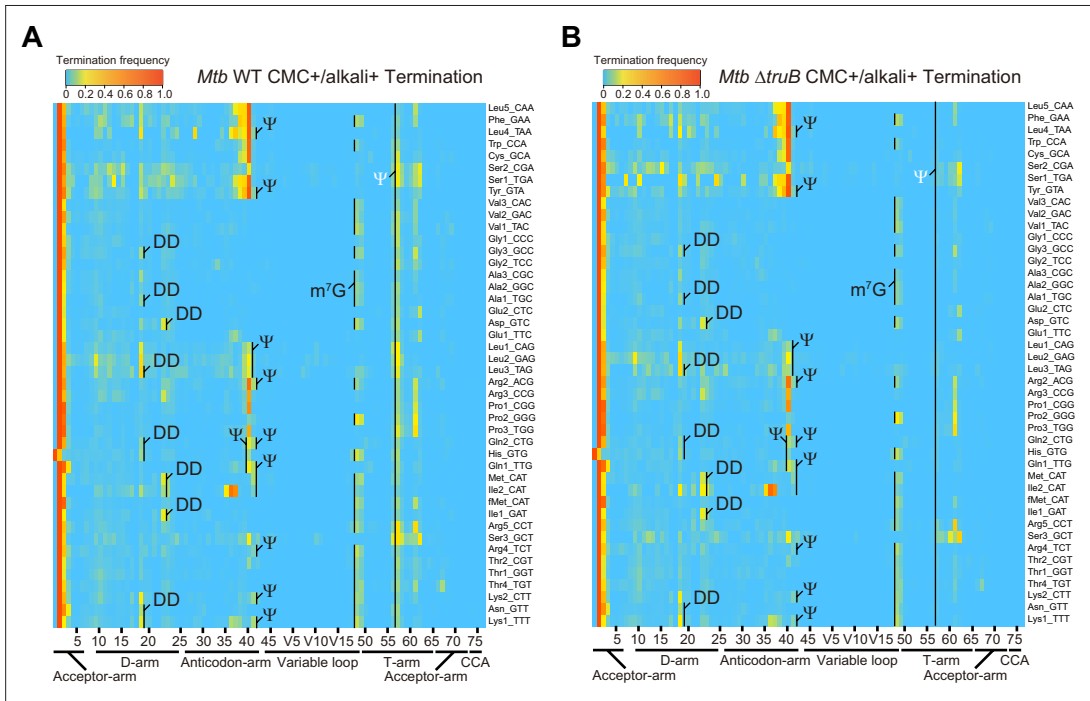

**Figure 6.** Heatmap of early termination frequency from sequencing of tRNAs isolated from wild-type (WT) and *MtbΔtruB* following 1-cyclohexyl-(2-morpholinoethyl) carbodiimide (CMC) treatment. Heatmap of early termination frequencies across tRNA molecules and positions for WT (left) and *MtbΔtruB* (right). Termination signals derived from position 55 are shown in white. The experiment was performed once.

The online version of this article includes the following source data and figure supplement(s) for figure 6:

**Source data 1.** Termination frequencies across tRNA molecules and positions for wild-type (WT) and *MtbΔtruB*.

**Figure supplement 1.** Misincorporation and termination signals derived from D at positions 20, 20A, and 21 in *Mtb* tRNAs.

**Figure supplement 1—source data 1.** Termination frequencies at positions 18 and 21, and misincorporation frequencies at positions 16 (B), 17, 20, and 20A to predict the presence of D in *Mtb* tRNAs.

**Figure supplement 2.** Misincorporation signals derived from m[7]G at position 46.

**Figure supplement 2—source data 1.** Termination frequencies at position 46 to predict the presence of m[7]G in *Mtb* tRNAs.

**Figure supplement 3.** Termination signals derived from pseudouridine at positions 55, 38, and 39.

**Figure supplement 3—source data 1.** Termination frequencies at positions 57, 40, and 41 to assess the pseudouridylation states at positions 55, 38, and 39 in *Mtb* tRNAs.

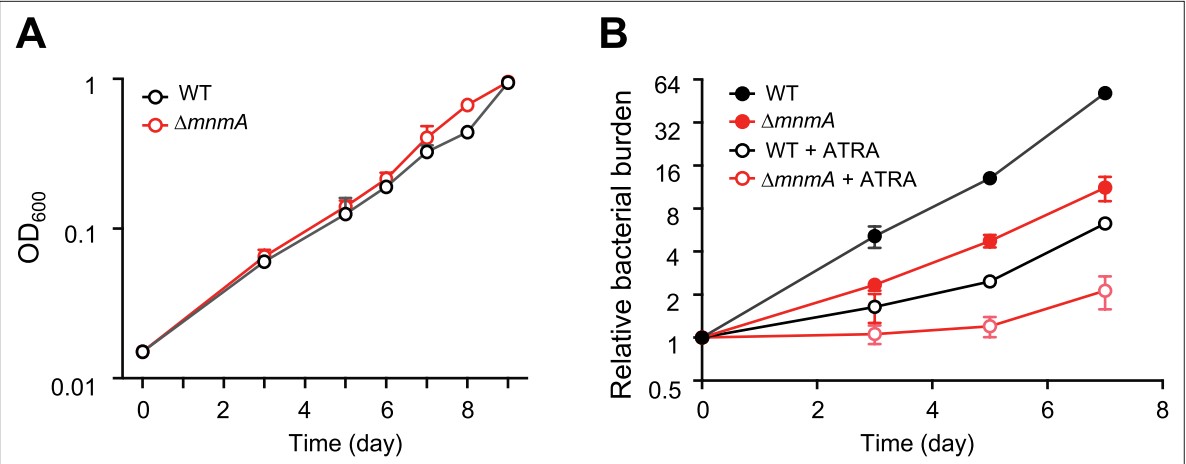

**Figure 7.** *MtbΔmnmA* is attenuated in a macrophage infection model. (**A**) Wild-type and *MtbΔmnmA* do not display growth differences in 7H9 medium. (**B**) Auto-luminescent wild-type and *ΔmnmA Mtb* strains were diluted to a multiplicity of infection of 2 bacteria per mouse bone marrow-derived macrophage with or without all-trans retinoic acid (ATRA). Survival was measured by luminescence and normalized to luminescence at time 0. Average values from three independent cultures (n=3) are shown with standard deviations.

The online version of this article includes the following source data for figure 7:

**Source data 1.** Mtb growth curve.

IAA treatment was developed for detecting $s^4U$ modification in pulse-chase experiments to measure RNA turnover (*Herzog et al., 2017*). We found that this treatment enhances the RT signatures of additional sulfur modifications as well, including $s^2C$ and $s^2U$, indicating that IAA should have general utility in tRNA-seq-based profiling of sulfur modifications, including in studies tracking changes in tRNA sulfuration (*Edwards et al., 2022*; *Laxman et al., 2013*).

Multiple modifications do not cause reverse transcriptase errors. Although tRNA-seq provides a simple and rapid method for profiling the landscape of modifications in all tRNA species, this approach does not enable comprehensive identification of modifications. Here, several modifications, including $t^6A$, $m^5U$, $mo^5U$, $m^5C$, Cm, Um, and Gm, predicted by the presence of *Mtb* homolog of tRNA modifying enzymes, such as TsaC2, TsaB, TsaD, TsaE, Trm2, TrcM, TrmR, TrmL, and TrmH, were not detected in tRNA-seq. Additionally, CMC treatment may not determine all Ψ positions, thus, the target of Rv3300 and Rv1540 remain unclear. Since these genes are similar to *E. coli rluA*, which also targets rRNA, these genes may target rRNAs instead of tRNAs. Mapping and further analysis of these modifications will require different approaches such as tRNA purification and RNA mass spectrometric analysis.

Several strong RT signatures were detected in *Mtb* tRNAs but not in *E. coli* (*Figure 2A*), including G45 in tRNA-Gly2. Most of these signals are strict misincorporations of C or T at A or G position, respectively. Since modifications at a purine position, for example, $m^1G$ and $m^1A$, generally cause random misincorporation of the other three nucleosides, these *Mtb*-specific misincorporation signatures are not likely to be derived from modification. Further mass spectrometric analysis of purified tRNAs will be necessary to assess whether these signals are indeed derived from *Mtb*-specific modifications; such *Mtb*-specific modifications could be related to *Mtb* pathogenesis and/or survival in the host.

The role of $s^2U$ in *Mtb* appears to be unusual. $s^2U$ is a universally conserved modification observed in all three domains of life at the wobble position in the anticodon. This modification enhances the stacking of $s^2U$ with U35 to stabilize the anticodon structure, facilitating codon–anticodon interactions. $s^2U$ is also recognized by multiple aminoacyl-tRNA synthetases for efficient amino acylation (*Giegé and Eriani, 2023*). Although elimination of this sulfur modification causes severe growth phenotypes in most organisms (*Kambampati and Lauhon, 2003*; *Dewez et al., 2008*), unexpectedly, the deletion of *mnmA* in *Mtb* did not attenuate growth of the pathogen in vitro, suggesting that the *Mtb* requirement for $s^2U$ modification differs from other organisms. The marked growth retardation of *ΔmnmA* strain within macrophages indicates the specific requirement of this modification within host cells. This

modification may be necessary for maintaining general translation efficiency inside host cells and/or facilitate the expression of specific genes that are necessary for survival within macrophages. In fact, lack of *mnmA* is reported to sensitize *E. coli* to oxidative stress, raising the possibility that s²U modification promotes *Mtb* growth under oxidative stress elicited by the host.

Differential codon usage between house-keeping genes and virulence genes could contribute to the differential growth phenotypes observed in vitro and in vivo in the Δ*mnmA* mutant. Among multiple codons encoding Glu, Gln, and Lys, s²U modification dependent codon usage might be preferentially distributed in genes associated with intracellular growth. For example, *Mtb* has two tRNA isoacceptors, tRNA-Glu1(s²UUC) and tRNA-Glu2(CUC), to decipher two Glu codons, GAA and GAG. According to the wobble paring rule, GAA is only decoded by tRNA-Glu1(s²UUC), whereas GAG is decoded by both tRNA-Glu1(s²UUC) and tRNA-Glu2(CUC); that is, GAG can be deciphered by an s²U-independent tRNA. Thus, genes required for intracellular growth might be enriched with GAA, an s²U-dependent codon. Similar codon usage differences could be present in Gln and Lys codons deciphered by s²U-containing tRNAs.

In most organisms, s²U is further modified into derivatives containing an additional chemical moiety at position 5 (*Björk and Hagervall, 2014*; *Karlsborn et al., 2014*; *Asano et al., 2018*). However, *Mtb* does not contain apparent homologs of the tRNA modifying enzymes that introduce the additional modifications to s²U. Thus, *Mtb* may contain s²U or s²U derivatives synthesized by other types of enzymes. Additional analyses to elucidate the structures of modified s²U in *Mtb* are warranted.

Another unexpected finding is the presence of a TrmR homolog in *Mtb*. TrmR in *B. subtilis* is a methylase that converts a 5-hydroxyuridine (ho⁵U) into a 5-methoxyuridine (mo⁵U) at the tRNA wobble position, suggesting that *Mtb* possesses mo⁵U. However, *Mtb* does not encode an apparent homolog for the enzyme that mediates the RNA hydroxylation reaction that yields ho⁵U, the substrate for TrmR. Although we cannot rule out the possibility that *Mtb* TrmR methylates a substrate other than RNA, it is possible that *Mtb* utilizes an unknown hydroxylation pathway for synthesizing ho⁵U.

Our findings serve as a valuable starting point for the research community to continue characterizing the physiological roles and mechanisms of *Mtb* tRNA modifications. Since tRNA-seq offers an efficient and scalable platform for surveying changes in tRNA modifications, this approach will be valuable across growth conditions, and may be extended to growth inside host cells. Finally, further studies elucidating the mechanisms by which tRNA modifications facilitate *Mtb* growth in host cells should be valuable for designing new therapeutics for tuberculosis.

## Materials and methods

**Key resources table**

| Reagent type (species) or resource | Designation | Source or reference | Identifiers | Additional information |
|---|---|---|---|---|
| Gene (*Mycobacterium tuberculosis*) | mnmA | NA | Uniprot: Rv3024c; Refseq: NP_217540.1 | |
| Gene (*Mycobacterium tuberculosis*) | truB | NA | Uniprot: Rv2793c; Refseq: NP_217309.1 | |
| Strain, strain background (*Mycobacterium tuberculosis*) | H37Rv | PMID:9634230 | | |
| Genetic reagent (*Mycobacterium tuberculosis*) | *Mtb*Δ*mnmA::zeo* | This paper | | Mtb H37Rv strain lacking *mnmA* |
| Genetic reagent (*Mycobacterium tuberculosis*) | *Mtb*Δ*truB::zeo* | This paper | | Mtb H37Rv strain lacking *truB* |
| Recombinant DNA reagent | pNit-RecET SacBR | NA | | For homologous recombination |
| Chemical compound | Iodoacetamide (IAA) | Sigma | | |
| Chemical compound | 1-Cyclohexyl-(2-morpholinoethyl) carbodiimide (CMC) | Sigma | | |

## Bacterial strains and growth conditions

*Mtb* strains were grown from frozen stocks into Middlebrook 7H9 medium supplemented with 0.2% glycerol, 0.05% Tween-80, and ADC (5 g/l bovine serum albumin, 2 g/l dextrose, 3 µg/ml catalase). Cultures were incubated at 37°C. Strains were grown to mid-log phase for all experiments (OD$_{600}$ 0.4–0.6). Growth was measured on a BioTek plate reader for in vitro growth by measuring OD$_{600}$ every 24 hr.

## Bacterial strain construction

*Supplementary file 4* depicts the strains, plasmids, primers, and recombinant DNA used for this study. Plasmids were built by restriction digest of a parental vector and inserts were prepared by Gibson assembly (*Gibson et al., 2009*). Plasmids were isolated from *E. coli* and confirmed via Sanger sequencing carried out by Genewiz, LLC (Massachusetts, USA).

### Deletion mutants

The knockout strain *MtbΔmnmA*::zeo (zeocin) was built using double-stranded recombineering in the parental *Mtb* strain H37Rv. A linear dsDNA fragment was constructed using stitch polymerase chain reaction (PCR) with the primers listed in *Supplementary file 4* which consisted of a 500-bp region upstream of *mnmA* (Rv3024c), 500-bp downstream region, and a *lox-zeo-lox* fragment. This cassette was transformed into an H37Rv recombineering strain as described (*Murphy et al., 2015*) and plated on 7H10 + zeocin plates.

## Homology search

Local BLAST was performed to search for *Mtb* homologs of tRNA modifying enzymes. First, the uniport IDs of tRNA modifying enzymes were obtained from Modomics (*Boccaletto et al., 2022*), and 12 proteins were manually added to the list (*Cho et al., 2023*; *Kimura et al., 2020*; *Sakai et al., 2019*; *Kimura et al., 2022*; *Takakura et al., 2019*; *Sakai et al., 2016*), including Q47319/TapT, P24188/TrhO, P76403/TrhP, O32034/TrhP1, O32035/TrhP2, P36566/CmoM, and Q87K36/TrcP, O34614/MnmM, Q8N5C7/DTWD1, Q8NBA8/DTWD2, O32036/TrmR, Q9KV41/AcpA. Uniprot ID provides a fasta file of tRNA modifying enzymes from the Uniprot database (*UniProt, 2023*). A blast database file was generated by 'makeblastdb' script using a fasta file of *Mtb* proteins (H37Rv strain) retrieved from NCBI. Then, the homologs of tRNA modifying enzymes were searched against the *Mtb* protein database using the fasta file of tRNA modifying enzymes as a query. Output format is defined by the following script: -outfmt '7 qacc sacc stitle score qcovs evalue pident' -evalue 1e-10. The output file was modified by excel (*Supplementary file 4*).

## Phylogenetic analysis of *Mtb* tRNA modifying enzyme homologs

Local BLAST was conducted to search for homologs of *Mtb* tRNA modifying enzymes in 120 manually picked organisms across three domains of life. A custom database was generated by combining the fasta files of organisms' proteins retrieved from NCBI. Homologs of 18 *Mtb* tRNA modifying enzymes were searched against the custom protein database using local blast. Log10 *E*-values were visualized by iTol (*Letunic and Bork, 2021*) with a phylogenetic tree generated by phyloT (*PhyloT, 2015*).

## tRNA sequencing
### Extraction of total RNA

Strains were grown to mid-log phase with the appropriate antibiotics and inducing agents described above. RNA was collected at the same OD$_{600}$ for each strain (between 0.4 and .6). Cells were left on ice for 20 min, then pelleted by centrifuging at 4000 rpm for 10 min at 4°C. Pellets were resuspended in 0.5–1 ml of TriZol (Life Technologies) and lysed using a BeadBug microtube homogenizer (Millipore Sigma). 200 µl of chloroform was added to each tube, after which samples obtained from *Mtb* strains were removed from biosafety level 3 precautions. Samples were centrifuged at 15,000 rpm for 15 min at 4°C and the aqueous layer was collected into a fresh tube. To the original tube, 250 µl of sodium acetate buffer (300 mM sodium acetate pH 5.2 and 10 mM ethylenediaminetetraacetic acid (EDTA) pH 8.0) was added, and samples were vortexed at 4°C for 5 min then centrifuged at 15,000 × *g* for 15 min at 4°C. The aqueous layer was added to the fresh sample-containing tubes. 400 µl chloroform

was added, and tubes were briefly vortexed and then centrifuged at 15,000 rpm for 1 min at 4°C. The aqueous phase was collected into a fresh tube and RNA recovered by ethanol precipitation. RNA pellets were resuspended in 10 mM sodium acetate pH 5.2 and stored at −80°C until processed for sequencing. Total RNA samples were alkali treated prior to tRNA extraction to deacylate all tRNAs (1 hr at 37°C in 100 mM Tris–HCl pH 9.0).

## Isolation of tRNA fraction
1–2 µg of total RNA was run on a 10% TBE-UREA gel (Thermo Fisher Scientific) at 250 V for 1 hr. Gels were stained with SYBR Gold (Thermo Fisher Scientific), and tRNA was excised. Excised gels containing tRNA fractions were mashed in RNAse-free tubes, and 300 µl elution buffer (300 mM NaOAc pH 5.5, 1 mM EDTA pH 8.0, 0.10% sodium dodecyl sulfate) was added to each tube. Samples were shaken on a thermoshaker (Eppendorf) for 1–4 hr at 37°C and supernatant was collected using an Ultrafree filter column (Millipore Sigma). tRNA was recovered by isopropanol precipitation.

## tRNA dephosphorylation
tRNA was dephosphorylated using QuickCIP (New England BioLabs) according to manufacturer instructions, and tRNA was collected by phenol–chloroform extraction followed by isopropanol precipitation.

## IAA treatment
IAA treatment was performed as described (**Herzog et al., 2017**). Briefly, 500 ng of total RNA is combined with 10 mM of IAA, 50 mM $NaPO_4$ pH 8.0, and 50% dimethyl sulfoxide (DMSO) in a final volume of 50 µl. Reactions were incubated at 50°C for 15 min and quenched with dithiothreitol (DTT).

## CMC treatment
CMC treatment was carried out as described in ref Briefly, 2.5 µg tRNA fraction in 0.5 µl was mixed with 15 µl CMC-BEU buffer with or without CMC (0.34 M or 0 M CMC, 7 M urea, 4 mM EDTA, and 50 mM bicine pH 7.9) and incubated at 37°C for 20 min. Adding 100 µl CMC stop solution (0.3 M NaOAc pH 5.2 and 100 mM EDTA) quenched the reaction. RNA was desalted with PD-10 desalting column (Cytiva) and recovered by ethanol precipitation. RNA was dissolved in 40 µl of 50 mM sodium carbonate buffer (pH 10.4) and incubated at 37°C for 4 hr, followed by ethanol precipitation.

## Adapter ligation
0.5 µL RNase inhibitor was added to 3.5 µl dephosphorylated tRNA (200–250 ng tRNA) and samples were boiled at 80°C for 2 min. Boiled tRNA was mixed with 12 µl PEG buffer mix (10 µl 50% PEG8000, 2 µl 10× buffer B0216S; New England Biolabs). 3 µl of 5′ adenylated linkers (**Supplementary file 4**) were added (33 pmol/µl) along with 1 µl T4 RNA ligase 2 truncated (New England BioLabs) and incubated at 25°C for 2.5 hr. Samples were recovered by isopropanol precipitation and run on a 10% TBE-Urea PAGE gel for 40 min at 250 V. Ligated products were recovered by gel excision as described above.

## Reverse transcription
Identical quantities of samples with different adapter sequences were pooled for reverse transcription for a total of 200–250 ng tRNA. Reverse transcription was performed by combining 2.1 µl dephosphorylated tRNA with 100 mM Tris–HCl pH 7.5, 0.5 mM EDTA, 1.25 µM RT primer (**Supplementary file 4**), 450 mM NaCl, 5 mM $MgCl_2$, 5 mM DTT, 500 nM TGIRT (InGex), and 15% PEG8000 in a final volume of 9 µl. Samples were incubated at 25°C for 30 min, after which 1 µl 10 mM dNTPs (New England BioLabs) were added and reactions incubated at 60°C for 1 hr. 1.15 µl NaOH was added, and samples were boiled for 15 min and run on a 10% TBE Urea PAGE gel at 250 V for 1 hr. Reverse transcription products were excised from gels and cDNA recovered by isopropanol precipitation. Linear single-stranded cDNA was circularized using CircLigase II (Lucigen) in accordance with manufacturer instructions.

## PCR of tRNA libraries

PCR reactions were set up using HF Phusion according to the manufacturer's instructions using a universal reverse primer (**Supplementary file 4**) and a different index primer for each pool of samples. PCR reactions were aliquoted into 4 tubes and collected after 6, 8, 10, and 12 cycles. Samples were run on a Native TBE PAGE gel (Thermo Fisher Scientific) at 180 V for 50 min, and amplified products were cut from the same cycle for each sequencing run. Samples were recovered by gel excision and isopropanol precipitation.

## Sequencing

Sequencing was performed on a MiSeq instrument (Illumina) using 150 bp single end reads with a version 3, 150 cycle kit.

## Analysis

3′ linker sequences and two nucleotides at the 5′ end were trimmed using cutadapt and fastx-trimmer. Bowtie v1.2.2 was used with default settings to map reads to reference *Mtb* tRNA sequences retrieved from Mycobrowser (**Kapopoulou et al., 2011**; **Supplementary file 5**). Mpileup files were generated using samtools (samtools mpileup -l -A --ff 4 -x -B -q 0 -d 10000000). For analysis of termination frequencies, 5′ end termini of mapped reads were piled up using bedtools genomecov (option, -d -5 -ibam). The number of 5′ termini at each tRNA position was divided by the total number of mapped termini at that position plus all upstream (5′) positions.

## Macrophage infection

Auto-luminescent wild-type and $\Delta mnmA$ *Mtb* strains grown to the same $OD_{600}$ were pelleted by centrifugation and prepared in RPMI media by soft spinning as described (**Saito et al., 2017**). Briefly, cells were washed, pelleted, resuspended, and centrifuged at $121 \times g$, with the top half of the centrifuged supernatant used. Suspensions were diluted to a multiplicity of infection of 2 bacteria per mouse bone marrow-derived macrophage by determining the $OD_{600}$. Macrophages were infected for 6 hr, followed by a phosphate-buffered saline wash and addition of RPMI with or without ATRA. ATRA promotes macrophage control of *Mtb* infection (**Babunovic et al., 2022**) and was used to assess strain survival in an increasingly restricted macrophage environment. Survival was measured by luminescence in a BioTek plate reader and normalized to luminescence reads at time 0.

## Acknowledgements

We appreciate all the members of Waldor lab for fruitful discussion and comments on the manuscript. We also thank Gregory Babunovic for his valuable assistance setting up the macrophage infections used here. This work is supported by NIH/NIAID grants to MKW (R01AI-042347) and EJR (P01AI-095208), a Dean's Innovation Award from Harvard Medical School to EJR and the Howard Hughes Medical Institute (HHMI) to MKW.

## Additional information

### Funding

| Funder | Grant reference number | Author |
|---|---|---|
| National Institute of Allergy and Infectious Diseases | R01AI-042347 | Matthew K Waldor |
| National Institute of Allergy and Infectious Diseases | P01AI-095208 | Eric J Rubin |
| Harvard Medical School | Dean's Innovation Award | Eric J Rubin |
| Howard Hughes Medical Institute | MKW | Matthew K Waldor |

| Funder | Grant reference number | Author |
|--------|------------------------|--------|

The funders had no role in study design, data collection, and interpretation, or the decision to submit the work for publication.

## Author contributions

Francesca G Tomasi, Conceptualization, Data curation, Investigation, Visualization, Methodology, Writing – original draft, Writing – review and editing; Satoshi Kimura, Conceptualization, Data curation, Supervision, Investigation, Visualization, Methodology, Writing – review and editing; Eric J Rubin, Matthew K Waldor, Conceptualization, Supervision, Funding acquisition, Writing – review and editing

## Author ORCIDs

Satoshi Kimura http://orcid.org/0000-0003-3555-5877
Eric J Rubin http://orcid.org/0000-0001-5120-962X
Matthew K Waldor http://orcid.org/0000-0003-1843-7000

Reviewer #1 (Public Review): https://doi.org/10.7554/eLife.87146.3.sa1
Reviewer #2 (Public Review): https://doi.org/10.7554/eLife.87146.3.sa2
Reviewer #3 (Public Review): https://doi.org/10.7554/eLife.87146.3.sa3
Author Response https://doi.org/10.7554/eLife.87146.3.sa4

---

# Additional files

## Supplementary files

- Supplementary file 1. tRNA modifying enzymes used as queries for BLAST.
- Supplementary file 2. Exploration of tRNA modifying enzymes in *Mtb* by BLAST.
- Supplementary file 3. Predicted tRNA modifying enzymes in *Mtb*.
- Supplementary file 4. List of primers, plasmids, and strains.
- Supplementary file 5. *Mtb* tRNA sequences.
- MDAR checklist

## Data availability

The sequencing data reported in this paper have been deposited in the NCBI Gene expression omnibus https://www.ncbi.nlm.nih.gov/geo/ (accession code, GSE240200).

The following dataset was generated:

| Author(s) | Year | Dataset title | Dataset URL | Database and Identifier |
|-----------|------|---------------|-------------|-------------------------|
| Tomasi FG, Kimura S, Rubin EJ, Waldor MK | 2023 | Profiling tRNA modifications in Mycobacterium tuberculosis | https://www.ncbi.nlm.nih.gov/geo/query/acc.cgi?acc=GSE240200 | NCBI Gene Expression Omnibus, GSE240200 |

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
