## [Editor Report · eLife assessment]

This is a **valuable** addition to the literature as it helps us understand the role of tRNA modifying enzymes in Mycobacterium tuberculosis. By knocking out one of the enzymes, the authors **convincingly** demonstrate the importance of tRNA-modifying enzymes for intra-host growth of tubercle bacteria. Some of the claims regarding modification as well as the role in virulence could be strengthened through further bioinformatics and phylogenetic analyses as well as experimental approaches. The work will be of interest to microbiologists.

---

## [Referee Report · Reviewer #1 (Public Review)]

Tomasi et al. performed a combination of bioinformatic, next-generation tRNA sequencing experiments to predict the set of tRNA modifications and their corresponding genes in the tRNAs of the pathogenic bacteria Mycobacterium tuberculosis. Long known to be important for translation accuracy and efficiency, tRNA modifications are now emerging as having regulatory roles. However, the basic knowledge of the position and nature of the modifications present in a given organism is very sparse beyond a handful of model organisms. Studies that can generate the tRNA modification maps in different organisms along the tree of life are good starting points for further studies. The focus here on a major human pathogen that is studied by a large community raises the general interest of the study. Finally, deletion of the gene mnmA responsible for the insertion of s2U at position 34 revealed defects in growth in macrophage but in test tubes suggesting regulatory roles that will warrant further studies. The conclusions of the paper are mostly supported by the data but the partial nature of the bioinformatic analysis and absence of Mass-Spectrometry data make it incomplete. The authors do not take advantage of the Mass spec data that is published for Mycobacterium bovis (PMID: 27834374) to discuss what they find.

---

## [Referee Report · Reviewer #2 (Public Review)]

In this study, Tomasi et al identify a series of tRNA modifying enzymes from Mtb, show their function in the relevant tRNA modifications and by using at least one deleted strain for MnmA, they show the relevance of tRNA modification in intra-host survival and postulate their potential role in pathogenesis.

Conceptually it is a wonderful study, given that tRNA modifications are so fundamental to all life forms, showing their role in Mtb growth in the host is significant. However, the authors have not thoroughly analyzed the phenotype. The growth defect aspect or impact on pathogenesis needs to be adequately addressed.

- The authors show that ΔmnmA grows equally well in the in vitro cultures as the WT. However, they show attenuated growth in the macrophages. Is it because Glu1_TTC and Gln1-TTG tRNAs are not the preferred tRNAs for incorporation of Glu and Gln, respectively? And for some reason, they get preferred over the alternate tRNAs during infection? What dictates this selectivity?

- As such the growth defect shown in macrophages would be more convincing if the authors also show the phenotype of complementation with WT mnmA.

An important consideration here is the universal nature of these modifications across the life forms. Any strategy to utilize these enzymes as the potential therapeutic candidate would have to factor in this important aspect.

---

## [Referee Report · Reviewer #3 (Public Review)]

The work presented in the manuscript tries to identify tRNA modifications present in Mycobacterium tuberculosis (Mtb) using reverse transcription-derived error signatures with tRNA-seq. The study identified enzyme homologs and correlates them with presence of respective tRNA modifications in Mtb. The study used several chemical treatments (IAA and alkali treatment) to further enhance the reverse transcription signals and confirms the presence of modifications in the bases. tRNA modifications by two enzymes TruB and MnmA were established by doing tRNA-seq of respective deletion mutants. Ultimately, authors show that MnmA-dependent tRNA modification is important for intracellular growth of Mtb. Overall, this report identifies multiple tRNA modifications and discuss their implication in Mtb infection.

---

## [Author Response]

The following is the authors’ response to the original reviews.

We thank the editors and reviewers for their thoughtful consideration of our manuscript. Here, we addressed the reviewers’ points.

**Reviewer #1 (Public Review):**
Tomasi et al. performed a combination of bioinformatic, next-generation tRNA sequencing experiments to predict the set of tRNA modifications and their corresponding genes in the tRNAs of the pathogenic bacteria Mycobacterium tuberculosis. Long known to be important for translation accuracy and efficiency, tRNA modifications are now emerging as having regulatory roles. However, the basic knowledge of the position and nature of the modifications present in a given organism is very sparse beyond a handful of model organisms. Studies that can generate the tRNA modification maps in different organisms along the tree of life are good starting points for further studies. The focus here on a major human pathogen that is studied by a large community raises the general interest of the study. Finally, deletion of the gene mnmA responsible for the insertion of s2U at position 34 revealed defects in in growth in macrophage but in test tubes suggesting regulatory roles that will warrant further studies. The conclusions of the paper are mostly supported by the data but the partial nature of the bioinformatic analysis and absence of Mass-Spectrometry data make it incomplete. The authors do not take advantage of the Mass spec data that is published for Mycobacterium bovis (PMID: 27834374) to discuss what they find.1. The authors say they took a list of proteins involved in tRNA modifications from Modomics and added manually a few but we do not know the exact set of proteins that were used to search the M. mycobacterium genome.

Thank you for pointing out this issue. We added the complete list of proteins used for the BLAST query as Supplemental Table 1.

1. The absence of mnmGE genes in TB suggested that the xcm5U derivatives are absent. These are present in M. bovis (PMID: 27834374). Are the MnmEG gene found in M. bovis? If yes, then the authors should perform a phylogenetic distribution analysis in the Mycobacterial clade to see when they disappeared. If they are not present in M. bovis then maybe a non-orthologous set of enzymes do the same reaction and then the authors really do not know what modification is present or not at U34 without LC-MS. The exact same argument can be given for the xmo5U derivatives that are also found in M.bovis but not predicted by the authors in M. tuberculosis.

The reviewer raises a valid point. In M. bovis mnm5U and cmo5U derivatives were observed in LC-MS analysis. However, we did not identify candidate genes known to be involved in the biogenesis of mnm5U and cmo5U in the Mycobacteriaceae, including M. bovis and Mtb, suggesting that if these modifications are indeed present, they are not synthesized through canonical biogenesis pathways in this family. There are several examples where the same modification is generated by distinct modification enzymes (Kimura, 2021). These observations raise the interesting possibility that in the Mycobacteriaceae and most species in actinomycetota (except for Bifidobacterium, Corynebacterium and Rhodococcus species), major wobble modifications are generated by biosynthesis pathways that are distinct from those employed by well-characterized organisms. Future studies will examine this hypothesis.

1. Why is the Psi32 predicted by the authors because of the presence of the Rv3300c/Psu9 gene not detected by CMC-treated tRNA seq while the other Psi residues are? Members of this family can modify both rRNA and tRNA. So the presence of the gene does not guarantee the presence of the modification in tRNAs

Thank you very much for the careful read. We did not include RluA in the list of query proteins because it is not classified as a tRNA modification enzyme in Modomics. Additionally, the CMC-coupled tRNA-seq is imperfect for detection of all pseudouridylated positions. Due to this limitation, we only assigned modifications that are both predicted by the presence of putative biosynthetic enzymes and RT-derived signatures. As the reviewer points out, we cannot rule out that this homolog targets only rRNAs. We clarified this possibility in the revised manuscript by adding the following sentence: “Additionally, CMC treatment may not identify Ψ at all positions, thus, the targets of Rv3300 and Rv1540 remain unclear. Since these genes are similar to *E. coli* rluA, which also targets rRNA, these genes may target rRNAs instead of tRNAs” (lines 298-300)

In the revised manuscript, RluA was added to the BLAST query for creating Fig. 2. Interestingly, Rv3300c is more similar to Pus9 than RluA, while Rv1540 is the Mtb gene most similar to *E. coli* RluA suggesting that these two genes encode pseudouridylases that target different species of tRNAs/rRNAs.

1. What are tsaBED not essential but tsaC (called sua5 by the authors) essential?

Thank you for pointing out this interesting observation. We are also curious about differences in the essentiality among t6A biogenesis genes. We speculate that TsaC has critical roles in cell viability other than t6A synthesis. TsaC synthesizes threonylcarbamoyl-AMP as an intermediate for t6A biogenesis. Thus, it is possible that this intermediate has a role in other essential cellular activities besides t6A biogenesis. Further study of these factors in Mtb could reveal interesting crosstalk between modification synthesis and other cellular activities.

**Reviewer #2 (Public Review):**
In this study, Tomasi et al identify a series of tRNA modifying enzymes from Mtb, show their function in the relevant tRNA modifications and by using at least one deleted strain for MnmA, they show the relevance of tRNA modification in intra-host survival and postulate their potential role in pathogenesis.Conceptually it is a wonderful study, given that tRNA modifications are so fundamental to all life forms, showing their role in Mtb growth in the host is significant. However, the authors have not thoroughly analyzed the phenotype. The growth defect aspect or impact on pathogenesis needs to be adequately addressed.The authors show that ΔmnmA grows equally well in the in vitro cultures as the WT. However, they show attenuated growth in the macrophages. Is it because Glu1_TTC and Gln1-TTG tRNAs are not the preferred tRNAs for incorporation of Glu and Gln, respectively? And for some reason, they get preferred over the alternate tRNAs during infection? What dictates this selectivity?

Thank you very much for raising this excellent point. As the reviewer suggests, the attenuation of ΔmnmA Mtb growth inside of macrophages could be caused by disparate codon usage between genes required for in vitro growth and intracellular growth. Among multiple codons encoding Glu, Gln, or Lys, s2U modification-dependent codons might be preferentially distributed in genes associated with intracellular growth. For example, Mtb has two tRNA isoacceptors, Glu1_TTC and Glu2_CTC, to decipher two Glu codons, GAA and GAG. According to the wobble pairing rule, GAA is only decoded by Glu1_TTC, whereas GAG is decoded by both Glu1_TTC and Glu2_CTC; i.e., GAG can be deciphered by an s2U-independent tRNA. Thus, genes required for intracellular growth might be enriched with GAA, an s2U-dependent codon. Similar codon usage differences could be present in Gln and Lys codons deciphered by s2U-containing tRNAs. In the revised manuscript, we included a new paragraph in the discussion explaining the possibility that differences in codon usage could contribute to the intracellular fitness defect of the ΔmnmA Mtb mutant (lines 323-332).

As such the growth defect shown in macrophages would be more convincing if the authors also show the phenotype of complementation with WT mnmA.

The reviewer raises a valid point. We note however, that Rv3023c, a putative transposase, is downstream of MnmA and unlike MnmA, Rv3023c appears to be dispensable for in vivo growth, according to the Tn-seq database (reference 44 and 45). Therefore, it is likely that the intracellular growth defect is caused by loss of mnmA.

An important consideration here is the universal nature of these modifications across the life forms. Any strategy to utilize these enzymes as the potential therapeutic candidate would have to factor in this important aspect.

This is a valid point. Targeting a pathogen-specific system enables avoidance of the adverse side effects caused by many therapeutic reagents. There are a couple of Mtb modification enzymes that are specific to bacteria and critical for Mtb fitness (e.g., TilS). These enzymes represent ideal potential therapeutic targets to impede Mtb intracellular growth.

**Reviewer #3 (Public Review):**
The work presented in the manuscript tries to identify tRNA modifications present in Mycobacterium tuberculosis (Mtb) using reverse transcription-derived error signatures with tRNA-seq. The study identified enzyme homologs and correlates them with presence of respective tRNA modifications in Mtb. The study used several chemical treatments (IAA and alkali treatment) to further enhance the reverse transcription signals and confirms the presence of modifications in the bases. tRNA modifications by two enzymes TruB and MnmA were established by doing tRNA-seq of respective deletion mutants. Ultimately, authors show that MnmA-dependent tRNA modification is important for intracellular growth of Mtb. Overall, this report identifies multiple tRNA modifications and discuss their implication in Mtb infection.Important points to be considered:The presence of tRNA-based modifications is well characterised across life forms including genus Mycobacterium (Mycobacterium tuberculosis: Varshney et al, NAR, 2004; Mycobacterium bovis: Chionh et al, Nat Commun, 2016; Mycobacterium abscessus: Thomas et al, NAR, 2020). These modifications are shown to be essential for pathogenesis of multiple organisms. A comparison of tRNA modification and their respective enzymes with host organism as well as other mycobacterium strains is required. This can be discussed in detail to understand the role of common as well as specific tRNA modifications implicated in pathogenesis.

The reviewer raises a fair point. However, with the exception of Chionh et al., the other studies cited here are not genome-wide characterization of tRNA modification. Re-analysis showed that the distribution of the tRNA modifying enzymes are very similar across mycobacterium strains, e.g., Mycobacterium smegmatis, Mycobacterium tuberculosis, and Mycobacterium abscessus, suggesting that modifications related to pathogenesis in Mtb may have different physiological roles in other Mycobacterium species. We included the distribution of tRNA modification enzymes across multiple mycobacterium species in a revised Fig. 1.

Authors state in line 293 "Several strong signatures were detected in Mtb tRNAs but not in *E. coli*". Authors can elaborate more on the unique features identified and their relevance in Mtb infection in the discussion or result section.

Thank you for the suggestion. However, the identity of these RT signatures and the relevance of these modifications for Mtb pathogenicity remains speculative at this point.

Deletion of MnmA is shown to be essential for *E. coli* growth under oxidative stress (Zhao et al, NAR, 2021). In similar lines, MnmA deleted Mtb suffers to grow in macrophage. Is oxidative stress in macrophage responsible for slow Mtb growth?

This is an excellent hypothesis which we have added to the revised manuscript (lines 320-322). “In fact, the absence of mnmA is reported to sensitize *E. coli* to oxidative stress, raising the possibility that s2U modification promotes Mtb growth under oxidative stress elicited by the host.”

Authors state in line 311-312 "Mtb does not contain apparent homologs of the tRNA modifying enzymes that introduce the additional modifications to s2U". This can be characterised further to rule out the possibility of other enzyme specifically employed by Mtb to introduce additional modification.

The reviewer raises a valid point. As discussed above (Reviewer #1, pt 2), Mtb may employ distinct enzymes to generate certain tRNA modifications. Future mass spec-based analyses of Mtb tRNAs will be carried out to identify the precise chemical structure of the sulfurated uridine, and subsequent studies will attempt to determine the enzymes that account for the biogenesis of these modifications.

Kimura, S. (2021). Distinct evolutionary pathways for the synthesis and function of tRNA modifications. Brief Funct Genomics, 20(2), 125-134. doi:10.1093/bfgp/elaa027

Reviewer #1 (Recommendations For The Authors):Additional data and AnalysesThe Modomics database is far from complete so it would be more rigorous to give the full set of genes that was used to do the searches as supplemental data.

Thank you for the suggestion. We added the list of the query genes as Supplemental Table 1.Minor points to be fixed

1. The authors name the psi32 synthase Rv3300c Pus9 when it is a member of the RluA family. It is not clear why the yeast/eukaryotic name was used.

We included enzymes from diverse species in our query, including eukaryotic genes. Indeed, we found that Rv3300c showed the lowest E-value among our query genes, therefore, we name Rv3300c as Pus9.

1. The sua5 gene name was used it should be tsaC to follow the accepted nomenclature.

We renamed Sua5 to TsaC2.

1. The statement lines 203-296 was totally unclear. I did not understand what the authors were trying to say at all.

This paragraph described how sequence context can result in different reverse transcription-derived signatures from dihydrouridine (D). We added a schematics describing this paragraph as Supplemental Fig. 6.

1. In reference, names with special characters should be fixed such as Börk.

We fixed the names with special characters.

**Reviewer #2 (Recommendations For The Authors):**
The authors state that at least some of tRNA modifying enzymes, while redundant for growth in vitro, may play a role during growth inside the macrophages, mostly due to the diverse stresses they could encounter.

We added a sentence, “In fact, the absence of mnmA is reported to sensitize *E. coli* to oxidative stress, suggesting that s2U modification is required for Mtb growth under oxidative stress elicited by the host” in the discussion.

• Ideally, authors could have tested the impact of diverse intracellular stresses that Mtb encounters, like redox stress, nitrosative, pH or nutritional stress, to check whether any of these stresses cause in vitro growth defects in ΔmnmA strain.

Thank you for the suggestion. This point will be addressed in future experiments.

This would be a wonderful way to show that under stress, the essentiality of tRNA modification enzymes changes.
**Reviewer #3 (Recommendations For The Authors):**
• In general, the clarity of the presentation can be improved.• Authors state that "MiaA, is non-essential in *E. coli*, but apparently essential in Mtb". While MiaA is shown to be critical for the fitness and virulence of extraintestinal pathogenic E. coli (Fleming et al, NAR, 2022). This can be clarified.

We rephrase as follows: “Unexpectedly, one modifying enzyme, MiaA, is non-essential in *E. coli* grown in nutrient-rich medium, but apparently …”

• Line numbers 130-132 is a repetition of line numbers 103-105

We repeated these sentences because the same claim was deduced from different experiments, i.e., BLAST search and tRNAseq.

• Line number 228: The presence of U at position 55 in the tRNAs can be included in the text for a better understanding.

We changed the text as following: “… Termination signatures derived from position 55, which is exclusively uridine in all tRNA species, increased in most tRNA species, suggesting …”

• A detailed pictorial depiction on comparing the modifications and enzymes from *E. coli* and Mtb can be included for easy understanding.

We created an *E. coli* tRNA modification map in the same format as Figure 2C and added it to the revised manuscript as a new Supplementary Fig. 1.